# Endoscopic transpapillary gallbladder drainage for the management of acute calculus cholecystitis patients unfit for urgent cholecystectomy

**Tae Hyeon Kim[1], Dong Eun Park[2], Hyung Ku Chon**[1] *

**1** Department of Internal Medicine, Wonkwang University College of Medicine and Hospital, Iksan, Republic of Korea, **2** Department of Surgery, Wonkwang University College of Medicine and Hospital, Iksan, Republic of Korea

* gipb2592@wku.ac.kr

## Abstract

**Data Availability Statement:** All relevant data are within the paper.

**Funding:** This report was supported by the Wonkwang University 2020.

### Objectives

Endoscopic transpapillary gallbladder drainage (ETGBD) has been proposed as an alternative to surgery or percutaneous cholecystostomy in patients with acute calculus cholecystitis (ACC). We aimed to evaluate the safety and efficacy of ETGBD via endoscopic transpapillary gallbladder stenting (ETGBS) or endoscopic naso-gallbladder drainage (ENGBD) as either a bridging or a definitive treatment option for patients with ACC when a cholecystectomy is delayed or cannot be performed.

### Methods

From July 2014 to December 2018, 171 patients with ACC in whom ETGBD were attempted were retrospectively reviewed. The technical and clinical success rates and adverse events were evaluated. Moreover, the predictive factors for technical success and the stent patency in the ETGBS group with high surgical risk were examined.

### Results

The technical and clinical success rates by intention-to-treat analysis for ETGBD were 90.6% (155/171) and 90.1% (154/171), respectively. Visible cystic duct on cholangiography were significant technical success predictor (adjusted odds ratio: 7.099, 95% confidence interval: 1.983–25.407, $P$ = 0.003) as per logistic regression analysis. Adverse events occurred in 12.2% of patients (21/171: mild pancreatitis, n = 9; acute cholangitis, n = 6; post-endoscopic sphincterotomy bleeding, n = 4; and stent migration, n = 1; ACC recurrence, n = 1), but all patients were treated with conservative management and endoscopic treatment. Among the ETGBS group, the median stent patency in 70 patients with high surgical risk was 503 days (interquartile range: 404.25–775 days).

**Competing interests:** The authors have declared that no competing interests exist.

**Abbreviations:** ACC, Acute calculus cholecystitis; GBD, Gallbladder drainage; PC, Percutaneous cholecystostomy; EUS, Endoscopic ultrasound; ETGBD, Endoscopic transpapillary gallbladder drainage; ERCP, Endoscopic retrograde cholangiography; LAMS, Lumen apposing metal stents; ETGBS, Endoscopic transpapillary gallbladder stenting; ENGBD, Endoscopic naso-gallbladder drainage; GB, gallbladder; CT, Computed tomography; MRI, Magnetic resonance imaging; CBD, Common bile duct; SpyDS, Spyglass™ DS Direct Visualization system; ASA, American Society of Anesthesiologists; EST, Endoscopic sphincterotomy.

## Conclusions

ETGBD, using either ETGBS or ENGBD, may be a suitable bridging option for ACC patients unfit for urgent cholecystectomy. In high surgical risk patients, ETGBS may be a promising and useful treatment modality with low ACC recurrence.

## Introduction

Cholecystectomy is the gold standard treatment for acute calculus cholecystitis (ACC) [1]. Early laparoscopic cholecystectomy for low-risk patients is safe and cost-effective [2, 3]. However, in high-risk patients, such as the elderly, critically ill, or those with severe comorbidities, cholecystectomy-related morbidity and mortality markedly increased [4, 5]. Poor surgical candidates may benefit from gallbladder drainage (GBD) with concomitant antibiotic treatment. Percutaneous cholecystostomy (PC) is a useful and widely available GBD method that is an alternative for patients in whom urgent surgery is contraindicated. PC has a high clinical success rate, ranging from 95% to 100% [6]. However, PC-related adverse events including bile leakage, hemorrhage, pneumothorax, inadvertent catheter dislodgement, and patient discomfort may occur in up to 25% of the patients [6, 7]. Additionally, the ACC recurrence rates have been reported to be 22–47% after PC catheter removal, without subsequent cholecystectomy [8, 9].

The effectiveness of endoscopic GBD for the treatment of patients with ACC for whom either surgery or PC is contraindicated has been investigated. There are two endoscopic GBD methods: the transmural approach guided by endoscopic ultrasound (EUS), and endoscopic transpapillary gallbladder drainage (ETGBD) under endoscopic retrograde cholangiopancreatography (ERCP). EUS-GBD was first described in 2007 [10]. The technical and clinical success rates were reported to be 84.6–100% and 86.7–100%, respectively [11, 12]. The complication rates associated with EUS-GBD ranged from 0–50% in various studies, which mostly reported in endoscopic experts [13–15]. Procedural complications have decreased with the use of dedicated devices and newly developed metal stents [16]. Recently, lumen apposing metal stents (LAMS), which have saddle-shaped design with flared ends and wide inner lumen, have been used for EUS-GBD and may reduce the risk of procedure-related complications, including stent migration or bile leakage. In a recent systematic review, the technical success rate, clinical success rate, and adverse events rate of EUS-GBD with a LAMS was 95.2%, 96.7%, and 8.8%, respectively [16, 17]. Nevertheless, interventional endoscopic experts are required, and the relatively high procedural costs may be challenging. Moreover, technical failure and severe adverse events, such as gut perforation or bile peritonitis that may lead to sepsis or death, occur even in expert hands. In addition, the feasibility and safety of interval cholecystectomy in patients who underwent EUS-GBD with a LAMS are not well reported, and there may be concerns due to severe inflammation or adhesion surrounding the gallbladder.

ETGBD consists of endoscopic transpapillary gallbladder stenting (ETGBS) and endoscopic nasobiliary gallbladder drainage (ENGBD) following standard selective bile duct cannulation with the use of ERCP. According to a systematic review, the technical and clinical success rates for ENGBD and ETGBS were 81–96% and 75–88%, respectively. Furthermore, the adverse event rates associated with both procedures were 3.6% and 6.3%, respectively [18]. However, previous studies were limited owing to small population sizes, and they are currently not widely used in the clinical practice. Studies evaluating the long-term outcomes of ETGBS in

patients unfit for surgery have also been limited. Therefore, we aimed to evaluate the safety and efficacy of ETGBD using ETGBS or ENGBD as either a bridge or a definitive treatment for patients with ACC when cholecystectomy is delayed or cannot be performed. We also examined the long-term ETGBS outcomes.

## Materials and methods

### Patients and definitions

This was a retrospective, single-center study that was conducted in accordance with the ethical guideline of the 1975 Declaration of Helsinki, and approved by Wonkwang University Hospital institutional review board (approval No. WKUH 2019-06-018). Given its retrospective nature, written informed consent to access the clinical data was not required by the board. From July 2014 to November 2018, patients with ACC who underwent attempted ERCP with ETGBD were enrolled, and their clinical data were retrospectively reviewed. All enrolled patients were diagnosed with ACC according to the Tokyo Guidelines [19]. Technical success was defined as the placement of one end of a pigtail stent into the gallbladder (GB) and the other end having been placed through the nose with a nasobiliary tube or within the duodenal lumen. Clinical success was defined as normalization of the ACC clinical hallmarks, including abdominal pain, fever, and leukocytosis, within 3 days of the intervention. Adverse events were defined as any complications during or after the procedure. Early adverse events included complications within 2 weeks of performing ETGBD and late adverse events included complications > 2 weeks after performing ETGBD. Stent patency was defined as the interval between performing ETGBS and stent dysfunction. Stent dysfunction was defined as a recurrent ACC or acute cholangitis occurrence. Acute cholangitis was defined as a new-onset Charcot's triad with biliary dilatation or stone on radiologic finding, as per Tokyo Guidelines [20].

### Endoscopic procedure and follow-up

ERCP was performed using a duodenoscope (JF 260V; Olympus Medical Systems, Tokyo, Japan) with the patient under sedation using pethidine and midazolam after obtaining informed consent. Cholangiography was performed after selective bile duct cannulation. Endoscopic sphinterotomy (EST) was performed in all cases except for a history of EST because of possible obstructive pancreatitis during a stent or naso-gallbladder tube insertion. Following cholangiography, a 0.025-inch angled hydrophilic guidewire (Visiglide®, Olympus Medical systems, Tokyo, Japan) was advanced through the cystic duct and into the GB. When the cystic duct was not visualized during cholangiography, a balloon occlusion cholangiogram of the distal common bile duct (CBD) was used to obtain the anatomy of the cystic duct. In case of an invisible cystic duct, we presumed the cystic duct entry by radiologic imaging and attempted to negotiate the guidewire into the cystic duct by quickly moving up and down with hydrophilic guidewire rotation preceding the catheter (Fig 1). When the guidewire advancement into the GB was challenging, a SpyGlass™ DS Direct Visualization system (SpyDS) (Boston Scientific, Natrick, Massachusetts, USA) was used to assist in the cystic duct cannulation. After successfully placing the guidewire into the GB, it was then coiled. A standard cannulation catheter was then carefully advanced into the GB via the cystic duct over the guidewire. When the standard cannulation catheter advancement into the cystic duct was challenging because of a tortuous and redundant cystic duct, a back and forth catheter movement could be used to straighten the cystic duct. In ETGBS, a 7-Fr 12–15 cm double pigtail plastic stent (Zimmon®, Cook medical, Bloomington, Indiana, USA) was deployed, with the proximal end in the GB and the distal end in the duodenum. In ENGBD, a 7-Fr or 5-Fr pigtail-type nasobiliary drainage tube (Liguory®, Cook medical, Bloomington, Indiana, USA) was placed

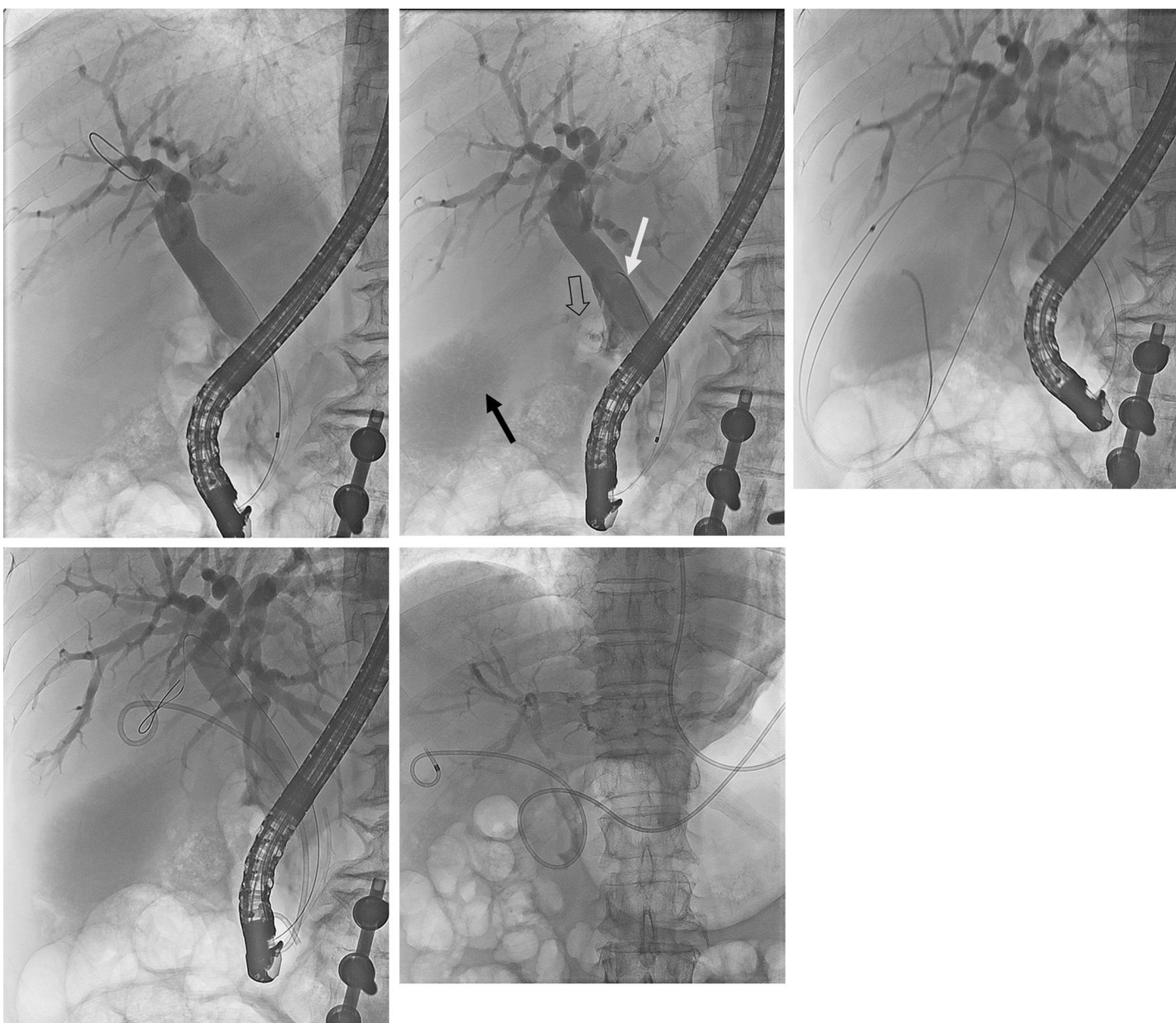

**Fig 1.** Endoscopic transpapillary gallbladder drainage via endoscopic transpapillary gallbladder stenting or endoscopic naso-gallbladder drainage A) Cholangiogram showing no visible cystic duct, B) Visible gallbladder (black arrow) and cystic duct (open arrow) after selective guidewire cannulation (white arrow) into the cystic duct, C) Coiled guidewire in the gallbladder, D) A 7-Fr 12 cm double pigtail stent is inserted between the gallbladder and duodenum, E) The 7-Fr pigtail-type nasobiliary drainage tube is inserted and left to indwell the gallbladder.

into the GB. Comcomitant bile duct stone was endeavored to remove at the same session. However, if the patient's condition was unstable, biliary plastic stent was inserted, and the bile duct stone was removed after stabilization of the patient. In case of ENGBD, naso-gallbladder catheter irrigation with saline was performed when the drained bile fluid was thick or with pus. In case of crossover from ENGBD to ETGBS, ENGBD catheter was removed following contrast injection through the catheter. A cystic duct patency was recognized and guidewire inserted into the GB. Subsequently, ETGBS procedure was performed in the same manner.

Two skilled endoscopists (THK and HKC) performed all procedures. Clinical follow-up assessment was conducted through retrospective chart reviews until December 31, 2019 or patient death, or by telephone for patients or their relatives not receiving routine care.

## Statistical analysis

Statistical analyses were performed using the SPSS 24.0 (IBM Corp., Armonk, New York, USA). Categorical data were analyzed using the Fisher's exact test or chi-squared test. Continuous variables were analyzed using the Student's t-test. The means and medians were used to summarize data for continuous variables. A $P$-value of $< 0.05$ was considered statistically significant. Stent patency was analyzed using the Kaplan-Meier estimator. A logistic regression analysis was performed to identify the predictive factors for successful ETGBD among the following variables: age, CBD diameter, presence of periampullary diverticulum, underwent magnetic resonance imaging (MRI) prior to the procedure, ACC severity, and cystic duct visible on cholangiography.

## Results

### Flow chart and baseline characteristics of the study population

During the study period, 171 patients with ACC underwent attempted ERCP with ETGBD because they were unfit for urgent cholecystectomy. A flow chart of the study cohort was shown in Fig 2. Of the 72 patients who underwent ENGBD, 60 received interval cholecystectomy after optimization of their comorbidities. Five patients with ENGBD refused cholecystectomy, and the ENGBD tube was removed after the inflammation improved. One patient who had lung cancer and undergoing chemotherapy died 12 days after the ENGBD due to pneumonia. ETGBS conversion was successfully performed in six patients with high surgical risk. Among those in the ETGBS success group (n = 83), interval cholecystectomy was performed in 10 patients and nine patients with simultaneous endoscopic retrograde biliary drainage using plastic stenting underwent ETGBS removal for remaining CBD stone treatment. The nine patients who underwent a second ERCP without additional ETGBS had no ACC recurrence during the study period. Seventy patients underwent follow-up including the remaining 64 patients as well as the six patients who underwent ETGBS to replace ENGBD. All patients with ETGBD failure (n = 16) were performed with PC. Among them, 10 patients underwent

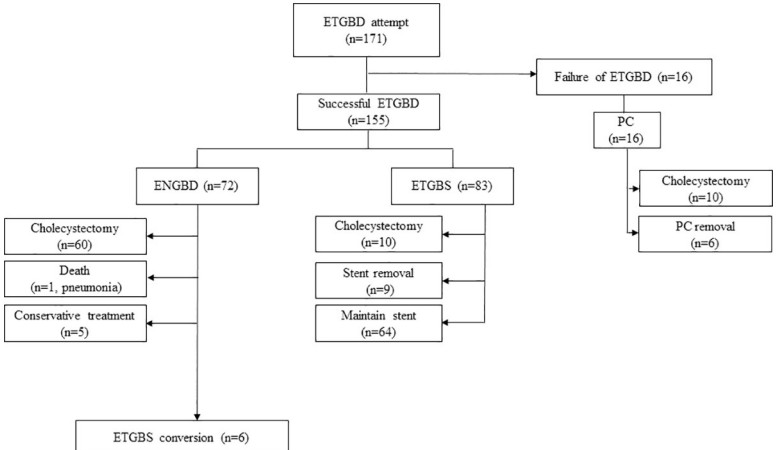

**Fig 2. Flow chart of the enrolled patients.**

interval cholecystectomy. Of the remaining six patients, two had recurrent ACC at 257 and 330 days post-PC removal and were treated with repeated PC.

The baseline characteristics of the enrolled patients are shown in Table 1. There was no significant difference between the ETGBD success and failure groups with respect to sex, age, whether interval cholecystectomy was performed, American Society of Anesthesiologists (ASA) class, underlying dementia, or ACC severity according to the Tokyo 2018 Guidelines. However, cystic duct visibility on cholangiography was significantly higher in the ETGBD success group (65.8% vs. 25%, $P = 0.001$).

## Procedural outcomes

Procedural outcomes and adverse events are summarized in Table 2. The technical and clinical success rates by intention-to-treat analysis were 90.6% (155/171) and 90.1% (154/171), respectively. Of the successful cases, eight had SpyDS-assisted cystic duct cannulation. ETGBD failed in 16 patients because the cystic duct could not be visualized (n = 12) or the catheter could not be advanced into the cystic duct (n = 4) due to cystic duct stricture. Among those in the ETGBD success group (n = 155), clinical symptoms with abnormal laboratory findings did not improve in one patient 3 days after the procedure, who subsequently underwent an interval cholecystectomy after PC.

The total adverse events rate was 12.2% (21/171). In early adverse events, the selective bile duct cannulation-related complications included post-endoscopic sphincterotomy (EST) bleeding (n = 4) or post-ERCP pancreatitis (n = 9). Early adverse events were treated using conservative management or endoscopic hemostasis. There were no procedure-related deaths.

The delayed adverse events included asymptomatic complete stent migration, ACC recurrence, and acute cholangitis. Complete stent migration occurred in one patient at 93 days after

**Table 1. Baseline characteristics of the study population (n = 171).**

|  |  | Success | Failure | P-value |
|---|---|---|---|---|
|  |  | (N = 155) | (N = 16) |  |
| Sex, male (%) |  | 83 (53.5) | 9 (56.3) | 0.493 |
| Age, mean (SD) |  | 72.88 (14.30) | 75.43(12.40) | 0.722 |
| Cholecystectomy, n (%) | No | 85 (54.8) | 10 (62.5) | 0.747 |
|  | Yes | 70 (45.2) | 6 (37.5) |  |
| ASA class, n (%) | I | 15 (9.7) | 2 (12.5) | 0.843 |
|  | II | 63 (40.6) | 5 (31.3) |  |
|  | III | 75 (48.4) | 9 (56.3) |  |
|  | IV | 2 (1.3) | 0 |  |
| Dementia, n (%) |  | 42 (27.1) | 5 (31.3) | 0.723 |
| Diagnosis | Presence of CBD stone with ACC, n (%) | 136 (87.7) | 15 (93.7) | 0.476 |
|  | ACC only, n (%) | 19 (12.3) | 1 (6.3) | 0.476 |
| Tokyo 2018 | G2: moderate, n (%) | 63 (40.6) | 4 (25.0) | 0.222 |
| Severity | G3: severe, n (%) | 92 (59.4) | 12 (75.0) |  |
| Previous EST, n (%) |  | 42 (27.1) | 5 (31.3) | 0.723 |
| Presence of PAD, n (%) |  | 62 (40.0) | 7 (43.8) | 0.771 |
| Visible cystic duct on cholangiography, n (%) |  | 102 (65.8) | 4 (25.0) | 0.001 |
| CBD diameter (mm), mean (SD) |  | 10.67 (3.64) | 11.43 (3.72) | 0.426 |
| Procedure time (min), SD |  | 16.3 (5.1) | 18.6 (2.5) | 0.086 |

SD standard deviation, ASA American society of anesthesiologists, CBD common bile duct, ACC acute calculus cholecystitis, G grade, EST endoscopic sphincterotomy, PAD periampullary diverticulum.

**Table 2. Outcomes of endoscopic transpapillary gallbladder drainage (n = 171).**

| | |
|---|---|
| **Technical success rate, % (n)** | **90.6 (155[$]/171)** |
| **Clinical success rate (by ITT), % (n)** | **90.1 (154/171)** |
| **Adverse events** | |
| Early adverse events | |
| Pancreatitis, % (n) | 5.2 (9) |
| Post EST bleeding, % (n) | 2.3 (4) |
| Late adverse events | |
| Stent migration, % (n) | 0.6 (1) |
| Acute calculus cholecystitis recurrence, % (n) | 0.6 (1) |
| Acute cholangitis, % (n) | 3.5 (6) |
| **Procedure-related mortality, n** | 0 |

[$]Eight cases used the Spyglass DS system-assisted cystic duct cannulation.

ITT Intention to treat analysis; EST endoscopic sphincterotomy.

ETGBS; however, there was no ACC recurrence until death due to pneumonia 61 days later. In one patient, ACC recurred at 43 days after the ETGBS, requiring an endoscopic reintervention. In six cases, acute cholangitis occurred at 150, 221, 299, 346, 381, and 399 days after the ETGBS and were managed with ERCP.

## Predictive factors for ETGBD success

Factors associated with the technical success of ETGBD were analyzed using logistic regression analysis. Age, CBD diameter, presence of periampullary diverticulum, MRI before the procedure, and ACC severity according to the Tokyo 2018 Guidelines had no significant correlation with the technical ETGBD success. However, a visible cystic duct on cholangiography was the strongest factor associated with the technical ETGBD success, as shown in Table 3 (adjusted odds ratio: 7.099, 95% confidence interval: 1.983–25.407, $P = 0.003$).

## Long-term ETGBS outcomes with high surgical risk

The baseline characteristics and long-term outcomes in 70 patients with ETGBS with high surgical risk are shown in Table 4. One patient had ACC recurrence (1.4%, 1/70) at 43 days and underwent reintervention. Six patients (8.5%, 6/70) developed acute cholangitis with choledocholithiasis at 150, 221, 299, 346, 381, and 399 days after the procedure, and were successfully

**Table 3. Logistic regression analysis of factors for successful endoscopic transpapillary gallbladder drainage.**

| Factor | Crude OR (95% CI) | P- value | Adjusted[a] OR (95% CI) | P- value |
|---|---|---|---|---|
| **Age** | 0.986 (0.948–1.026) | 0.491 | 0.969 (0.916–1.024) | 0.259 |
| **CBD diameter (mm)** | 0.945 (0.824–1.085) | 0.424 | 1.016 (0.853–1.209) | 0.863 |
| **PAD** | 0.857 (0.303–2.422) | 0.771 | 1.054 (0.310–3.581) | 0.933 |
| **MRCP** | 0.612 (0.216–1.738) | 0.357 | 0.467 (0.149–1.468) | 0.193 |
| **ACC severity according to Tokyo 2018 guideline** | 0.935 (0.436–2.005) | 0.862 | 0.831 (0.286–2.410) | 0.733 |
| **Visible cystic duct on cholangiography** | 5.774 (1.775–18.775) | 0.003 | 7.099 (1.983–25.407) | 0.003 |

OR odds ratio, CI confidence interval, CBD common bile duct, PAD periampullary diverticulum, MRCP magnetic resonance cholangiopancreatograph, ACC acute calculus cholecystitis.

[a]Adjusted for all the variables in the Table.

**Table 4. Baseline characteristics and long-term outcomes after endoscopic transpapillary gallbladder stenting for acute calculus cholecystitis in 70 patients with high surgical risks.**

| | | |
|---|---|---|
| Sex, male, n (%) | | 33 (46.5) |
| Age, mean (SD) | | 81.8 (6.8) |
| Dementia, n (%) | | 38 (53.5) |
| ASA class, n (%) | | |
| | II | 2 (2.8) |
| | III | 66 (94.4) |
| | IV | 2 (2.8) |
| Follow-up, mean (range), days | | 495.8 (43–1,117) |
| Recurrence of cholecystitis, n (%) | | 1 (1.4) |
| Re-intervention for acute cholecystitis, n (%) | | 1 (1.4) |
| Occurrence of cholangitis with choledocholithiasis, n (%) | | 6 (8.5) |
| Stent migration, n (%) | | 1 (1.4) |
| Patient status on follow-up, n (%) | | |
| | Alive | 50 (71.4) |
| | Dead | 20 (28.6) |

SD standard deviation, ASA American society of anesthesiologists.

retreated endoscopically. The median stent patency was 503 days (interquartile range: 404.25–775 days), as calculated using the Kaplan-Meier curve (Fig 3).

## Discussion

Early laparoscopic cholecystectomy is the preferred therapeutic option for ACC. However, GBD may be required as the initial treatment in patients for whom urgent cholecystectomy is contraindicated. PC is a widely used GBD procedure with high technical and clinical success. However, PC may result in a lower quality of life as well as post-procedural pain. In patients with high surgical risks, such as old age, long-term indwelling external drainage may be required; therefore, catheter-related infections or inadvertent tube dislodgements can be challenging. ETGBD including ENGBD or ETGBS was first reported 30 years ago [21]. The

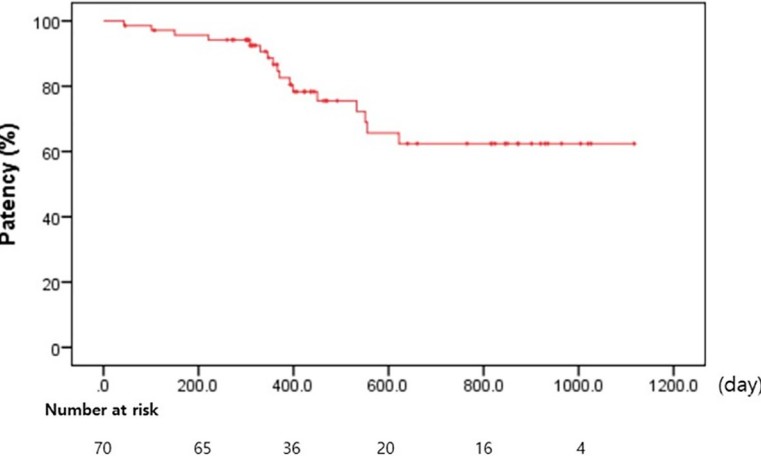

**Fig 3. Kaplan-Meier plot of the probability of cumulative biliary events for patients after endoscopic transpapillary gallbladder stenting.**

procedure was performed under ERCP, through which the GB was drained via the cystic duct using a nasobiliary tube or a double pigtail stent across the papilla. As per the Tokyo 2018 Guidelines, ETGBD is recommended as an alternative procedure to PC, especially in patients with severe coagulopathy, thrombocytopenia, or anatomically inaccessible location [22]. Compared with PC, ETGBD may provide more physiological drainage and be more cost-effective by shortening hospital stays, decreasing patient pain, and by being a more tolerable procedure [23]. As a bridging option to subsequent cholecystectomy, ETGBD had lower rates of unplanned repeated interventions compared with PC without affecting surgical outcomes [24]. In addition, concomitant CBD stones can be removed during the same session. However, ETGBD is not widely used due to the following technical challenges: (1) invisible cystic duct following cholangiography; (2) difficulty inserting guidewires into the GB via a tortuous and redundant cystic duct; (3) blocking of the guidewire, naso-gallbladder catheter, or pigtail stent advancement into the GB due to severe cystic stricture, or impacted stones in the cystic duct or in the GB neck; and (4) potential risk of cystic duct injury by guidewire or catheter manipulation.

We found that the technical and clinical success rates by intention-to-treat analysis for ETGBD were 90.6% and 90.1%, respectively. These outcomes were similar to, or higher than, other studies despite larger sample sizes [18]. Experienced endoscopic experts, skilled assistance, and the use of SpyDS may all be responsible for the high technical success. Under direct cholangioscopy-using SpyDS, the cystic duct entrance was easily detected and advancement of a guidewire or SpyDS scope into the cystic duct could be achieved, even in invisible cystic duct cases following cholangiography. In our study, Spy DS-assisted cystic duct cannulation was attempted in ten patients with high surgical risk and was successful in eight patients (S1 Table). Therefore, the technical success rate increased by 4.6%.

In the present study, most adverse events occurred in patients with naïve papilla, including post- ERCP pancreatitis (9/171, 5.2%) or post-EST hemorrhage (4/171, 2.3%). In 136 out of 155 patients with ETGBD success, concomitant CBD stone removal was successfully performed during the same session. Therefore, we suggest that ETGBD should be considered as a primary GBD method in ACC patients with both suspected acute cholangitis with choledocholithiasis and those for whom urgent surgery is contraindicated. In particular, the protocol for managing ACC in patients with a history of biliary sphincterotomy should consider including this approach.

Making decisions with ENGBD or ETGBS depends on the underlying disease, operability, or the nature of the aspirated GB fluid. ETGBS were performed in patients with high surgical risks or underlying dementia, whereas ENGBD were performed in operable patients. However, when thick purulent fluids were aspirated in patients with high surgical risks, a two-step approach was applied. ENGBD was initially performed. Subsequently, ETGBS conversion to replace ENGBD was performed after the inflammation improved with antibiotic therapy and topical irrigation with saline through the ENGBD tube. Due to our clinical approach, significant differences in baseline characteristics between the ENGBD and ETGBS groups were reported in our study (S2 Table). However, Yang et al. [25] detected no significant differences between ENGBD and ETGBS in terms of technical and clinical results in a prospective study designed to determine which ETGBD bridging method was clinically ideal before an interval cholecystectomy.

In the present study, stent dislodgement occurred in only one patient and ACC recurrence occurred in only one case in the ETGBS group with high surgical risks. Several studies have also reported that ETGBS may be a promising definitive treatment modality in patients with high surgical risks [26–29]. The exact mechanisms of these results are not yet fully understood, but a possible explanation could be that the anchoring system of the double pigtail stent

between the GB and duodenum with a narrow diameter of the cystic duct on the proximal end of the stent may have contributed to preventing stent migration. In addition, we hypothesized that GB atrophy after performing ETGBS may have occurred; therefore, the amount of bile flow from the GB to the duodenum may have decreased. Even if stent occlusion occurs, a small amount of bile flow can be maintained in the cystic duct adjacent to the stent. Moreover, stents prevent cystic duct clogging from the migratory GB stone; therefore, stent patency may be possible. However, additional functional and physiologic studies after long standing ETGBS are needed to clarify why the recurrence rates were low and why stent patency remained.

Lee et al. [29] reported that the median complication-free interval for ETGBS was 760 days. They suggested that observations for at least 2 years after ETGBS without stent removal or change might be needed. Several studies also demonstrated that there was no acute cholecysti- tis recurrence until 3 years after ETGBS in patients with liver cirrhosis or end-stage liver dis- ease awaiting orthotopic liver transplantation [30, 31]. Due to the short follow-up duration in the current study compared with other studies, the median stent patency was 503 days (inter- quartile range: 404.25–775 days). However, most of the patients were still under observation and we expect longer stent patency with longer follow-up periods.

In this study, six patients with ETGBS had cholangitis with choledocholithiasis. Theoreti- cally, a prolonged indwelling plastic stent in the bile duct may result in the formation of bile duct stones, which results from the calcium bilirubinate precipitation via promotion of bacte- rial proliferation, biofilm formation, and bacterial beta-glucuronidase release [32]. However, all patients who developed cholangitis underwent mechanical lithotripsy for concomitant bile duct stone removal during the same ETGBS session. Furthermore, three of these patients already had a cholangitis with choledocholithiasis history. Therefore, these patient factors may have also contributed to the cholangitis recurrence.

Recent studies comparing the EUS-GBD and ETGBD outcomes in patients with acute cho- lecystitis were published, and the authors reported that EUS-GBD resulted in higher technical and clinical success with lower cholecystitis recurrences than ETGBD [33, 34]. Conversely, these studies were conducted at tertiary care high-volume centers with EUS expertise. The sig- nificant differences in baseline characteristics of the study populations and the lack of report- ing follow-up durations with a defined post-procedural protocol, could have accounted for the observed outcome differences. Nevertheless, while recent studies mainly describe the EUS-GBD advantages, the ETGBD merits and importance are still emphasized. Additionally, controversies regarding EUS-GBD include: non-standardized skills compared with ETGBD through ERCP; highly complex interventions with steep learning curves [12, 35]; various metal stents used in EUS-GBD being substantially more expensive than plastic stents [36]; possibility of ERCP being still required in patients with choledocholithiasis after EUS-GBD; and possibil- ity of rescue therapy in failed EUS-GBD being challenging and creating serious complications that are likely to worsen the patient's condition. Higa et al. [33] reported that more adverse events with procedure-related deaths occurred in their EUS-GBD group. In patients with interval cholecystectomy, extra effort to repair the duodenal fistula during the surgery was required. However, in high-volume centers with skilled interventional endoscopists, EUS-GBD is useful as a first line treatment modality for the management of ACC in patients with high surgical risk, especially in those with malignancies involving the cystic duct or an indwelled metal stent covered with a cystic duct orifice [37]. In addition, if ETGBS failed as an initial method for GB decompression, EUS-GBD may be considered as a secondary option in patients with ACC who are unfit urgent or interval cholecystectomy.

In this study, several limitations were noted. First, this was a retrospective study with possi- ble selection bias and missed information. However, the data were prospectively collected and checked carefully. Secondly, this study was not designed to be compared with other gallbladder

drainage methods such as EUS-GBD or PC. Thirdly, we did not include any outcomes of costs/health economic analyses, which may be important factors to consider in the clinical practice. However, in one study [38] including cost-effectiveness analysis with PC, EUS-GBS, and ETGBD, the endoscopic approaches for AC are more cost-effective than PC. Moreover, ETGBD was favored over EUS-GBD.

In conclusion, ETGBD using either ETGBS or ENGBD may be a suitable option for ACC patients for whom urgent cholecystectomy are contraindicated. ETGBS has acceptable adverse events and may be a promising definitive treatment modality for high-risk surgical patients. However, additional investigations with well-designed multicenter prospective studies and long-term follow-ups are needed to validate our results.

## Supporting information

**S1 Table. Baseline characteristics of patients with successful SpyDS assisted cystic duct cannulation.**
(DOCX)

**S2 Table. Baseline characteristics of patients with endoscopic transpapillary gallbladder stenting (ETGBS) and endoscopic nasogallbladder drainage (ENGBD).**
(DOCX)

## Author Contributions

**Conceptualization:** Hyung Ku Chon.

**Formal analysis:** Hyung Ku Chon.

**Investigation:** Tae Hyeon Kim, Hyung Ku Chon.

**Methodology:** Hyung Ku Chon.

**Resources:** Tae Hyeon Kim, Dong Eun Park, Hyung Ku Chon.

**Supervision:** Tae Hyeon Kim, Dong Eun Park, Hyung Ku Chon.

**Writing – original draft:** Hyung Ku Chon.

**Writing – review & editing:** Tae Hyeon Kim, Hyung Ku Chon.

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
