## [Decision Letter · Decision Letter 0]

9 Aug 2020

PONE-D-20-19852

Endoscopic transpapillary gallbladder drainage for the management of acute calculus cholecystitis patients unfit for urgent cholecystectomy

PLOS ONE

Dear Dr. Chon,

Thank you for submitting your manuscript to PLOS ONE. After careful consideration, we feel that it has merit but does not fully meet PLOS ONE’s publication criteria as it currently stands. Therefore, we invite you to submit a revised version of the manuscript that addresses the points raised during the review process.

We look forward to receiving your revised manuscript.

Kind regards,

Ezio Lanza, M.D.

Academic Editor

PLOS ONE

2. In the ethics statement in the manuscript and in the online submission form, please provide additional information about the patient records used in your retrospective study. Specifically, please ensure that you have discussed whether all data were fully anonymized before you accessed them and/or whether the IRB or ethics committee waived the requirement for informed consent. If patients provided informed written consent to have data from their medical records used in research, please include this information. In addition, please include the dates upon which authors accessed the clinical data sources used in this study.

3. Please refer to any post-hoc corrections to correct for multiple comparisons during your statistical analyses. If these were not performed please justify the reasons. Please refer to our statistical reporting guidelines for assistance (https://journals.plos.org/plosone/s/submission-guidelines.#loc-statistical-reporting).

Reviewer #1: Dear Editor,

I read with interest the manuscript by Tae Hyeon Kim et al entitled “Endoscopic transpapillary gallbladder drainage for the management of acute calculus cholecystitis patients unfit for urgent cholecystectomy”

The work deals with a relevant and trending topic considering the impact of acute calculous cholecystitis on fragile patients. The work was conducted with a good methodology and it carries some interesting results; however, there are some points that I would like to stress.

Revision:

• Introduction: when the authors deal with the complication rates associated with EUS-GBD they refer to paper published between 2011 and 2014, a period in which lumen-apposing metal stent where not used. These data should be updated with more recent studies reported in several review published. Please include one of them such as :

• Flow chart and baseline characteristics of the study population, line 10: there is a grammatical error, “stentfig” instead of “stenting”.

The Authors reported ‘nine patients with simultaneous endoscopic retrograde biliary drainage using plastic stenting underwent ETGBS removal for remaining CBD stone treatment’. Why these patients are excluded in the subsequent follow-up? They had recurrence of ACC?Please specify.

• plastic stentfig underwent ETGBS removal for remaining CBD stone treatment.

• Procedural outcomes: the authors report that among the successful cases the SpyGlass system was used 8 times; have there also been cases of failed SpyGlass attempts?;

• Adverse events: acute calculous cholecystitis recurrence and acute cholangitis with choledocholithiasis occurred during the follow-up period were not counted among the adverse events; authors might motivate this choice;

• Discussion: as reported by the authors, endoscopic transpapillary gallbladder drainage could be a challenging technique with some pitfalls and it sometimes requires a two-step approach; could the authors comment on the possibility of performing several endoscopic procedures on the same fragile patient rather than just one drainage session (e.g. EUS-GBD)?;

Morevoer, the Authors included in this study 171 patients. A definitive endoscopic treatment was performed only in 70 patients. This is the main difference between the transpapillary approach and the EUS-GBD. As reported in a recent review on gallbladder drainage(PMID 32523368 ) in patients with AC who are not candidates for surgery, EUS-GBD represents a more appropriate treatment compared to the transpapillary approach. ETGBD can still be considered in patients with choledocholithiasis, as it allows concomitant stone removal and physiologic drainage. Please comment this topic in the discussion including this review in the manuscript.

• Discussion: in the last part of the discussion it is said that some minor procedural complications may have been missed; could the authors be more specific?.

Reviewer #2: This is a well designed retrospective analysis about endoscopic transpapillary GB drainage in acute calcolous colecistitis for patients unfit for urgent cholecistectomy.

In the LAMS (Lumen Apposing Metal stent) era it is crutial to discuss and evaluate an “anatomically natural” transpapillary approach in this setting of patients.

Please consider the following minor comments.

• In the Introduction section, following the sentence about LAMS “Moreover, technical failure and severe adverse events, such as gut perforation or bile peritonitis that may lead to sepsis or death, occur even in expert hands” I would add that in LAMS patients some concerns are raised in terms of feasibility and morbidity of interval cholecistectomy due to ahdesions between gallbladder and Stomach (or duodeum).

Because of that, if on one hand the EUS-guided gallbladder drainage with LAMS is gaining an important role for patients unfit for surgery, on the other hand it is still not proved it can be effectively used in patients who are possible candidates for delayed surgery

• In the material and Methods section, under the paragraph “endoscopic procedure and follow up” please edit the following sentence:

“In case of crossover from ENGBD to ETGBD, ENGBD catheter was removed […]” into the following “In case of crossover from ENGBD to ETGBS, ENGBD catheter was removed”

• About the patients who underwent SpyGlass assistance for cystic duct cannulation:

please specify the cases in whom Spy glass was used and if they were candidate for delayed surgery. Due to high costs this option maybe should be used in patient unfit for surgery with long stent/nasobiliary drainage indwelling time. In those patients for whom biliary drainage is a certain bridging option PT should be preferred in case of failure of standard cystic cannulation techniques

Even if you specified that a cost-effectiveness analysis has not been included in your study, a further description of this subpopulation of patients is beneficial to the clarity of the paper

---

## [Author Response · Author response to Decision Letter 0]

1 Sep 2020

Thank you for kind review and your comments concerning our manuscript.

I have replied to your important comments in a point-by-point manner. My reply comments are indicated in red.

I revised our manuscript as PLOS ONE’s style requirements.

2. In the ethics statement in the manuscript and in the online submission form, please provide additional information about the patient records used in your retrospective study. Specifically, please ensure that you have discussed whether all data were fully anonymized before you accessed them and/or whether the IRB or ethics committee waived the requirement for informed consent. If patients provided informed written consent to have data from their medical records used in research, please include this information. In addition, please include the dates upon which authors accessed the clinical data sources used in this study.

I added the revised sentences as follows: 

“This was a retrospective, single-center study that was conducted in accordance with the ethical guideline of the 1975 Declaration of Helsinki, and approved by our institutional review board (approval No. WKUH 2019-06-018). Given its retrospective nature, written informed consent to access the clinical data was not required by the board.” 

3. Please refer to any post-hoc corrections to correct for multiple comparisons during your statistical analyses. If these were not performed please justify the reasons. Please refer to our statistical reporting guidelines for assistance (https://journals.plos.org/plosone/s/submission-guidelines.#loc-statistical-reporting).

In our paper, categorical data were analyzed using the Fisher’s exact test or chi-squared test. Continuous variables were analyzed using the Student’s t-test. Thus, it would be unnecessary any post-hoc correction to correct for comparisons. 

Reviewer #1: Dear Editor,

I read with interest the manuscript by Tae Hyeon Kim et al entitled “Endoscopic transpapillary gallbladder drainage for the management of acute calculus cholecystitis patients unfit for urgent cholecystectomy”

The work deals with a relevant and trending topic considering the impact of acute calculous cholecystitis on fragile patients. The work was conducted with a good methodology and it carries some interesting results; however, there are some points that I would like to stress.

Revision

• Introduction: when the authors deal with the complication rates associated with EUS-GBD they refer to paper published between 2011 and 2014, a period in which lumen-apposing metal stent where not used. These data should be updated with more recent studies reported in several review published. Please include one of them such as:

As you mentioned, lumen apposing metal stents (LAMS) are recently used for EUS-GBD and procedure related complications including stent migration or bile leakage owing to lack of apposition of duodenal or gastric wall may be reduced. In a recent systematic review, technical success rate, clinical success rate, and adverse events rate of EUS-GBD with LAMS was 95.2%, 96.7%, and 8.8%, respectively. 

We added the sentences as follows: 

Recently, lumen apposing metal stents (LAMS), which have saddle-shaped design with flared ends and wide inner lumen, have been used for EUS-GBD and may reduce the risk of procedure-related complications, including stent migration or bile leakage. In a recent systematic review, the technical success rate, clinical success rate, and adverse events rate of EUS-GBD with a LAMS was 95.2%, 96.7%, and 8.8%, respectively [16, 17]. 

[Jain D, Bhandari BS, Agrawal N, Singhal S. Endoscopic Ultrasound-Guided Gallbladder Drainage Using a Lumen-Apposing Metal Stent for Acute Cholecystitis: A Systematic Review. Clin Endosc. 2018;51(5):450-62. doi: 10.5946/ce.2018.024. PubMed PMID: 29852730; PubMed Central PMCID: PMCPMC6182281.] 

[Kalva NR, Vanar V, Forcione D, Bechtold ML, Puli SR. Efficacy and Safety of Lumen Apposing Self-Expandable Metal Stents for EUS Guided Cholecystostomy: A Meta-Analysis and Systematic Review. Can J Gastroenterol Hepatol. 2018;2018:7070961. doi: 10.1155/2018/7070961. PubMed PMID: 29850458; PubMed Central PMCID: PMCPMC5925026.]

• Flow chart and baseline characteristics of the study population, line 10: there is a grammatical error, “stentfig” instead of “stenting”.

We corrected our grammatical error from “stentfig” to “stenting”.

The Authors reported ‘nine patients with simultaneous endoscopic retrograde biliary drainage using plastic stenting underwent ETGBS removal for remaining CBD stone treatment’. Why these patients are excluded in the subsequent follow-up? They had recurrence of ACC? Please specify.

In this study, the nine patients with remaining CBD stone with an initial ETGBS were performed ETGBS removal for CBD stone clearance via a second ERCP. We did not perform further ETGBS after CBD stone clearance because the gallbladder inflammation in those patients was improved and, to date, there was no evidence of efficacy of prophylactic ETGBS. However, the nine patients had no ACC recurrence during study period. We followed the all enrolled 171 patients during the study period. But, our flow chart may confused that only 70 patients with ETGBS conducted subsequent follow-up. Therefore, we added our revised the sentences and changed the flow chart as follows: The nine patients who underwent a second ERCP without additional ETGBS had no ACC recurrence during the study period. 

• plastic stentfig underwent ETGBS removal for remaining CBD stone treatment.

 We corrected our grammatical error from “stentfig” to “stenting”.

• Procedural outcomes: the authors report that among the successful cases the SpyGlass system was used 8 times; have there also been cases of failed SpyGlass attempts?;

We added the sentences as follows: In our study, Spy DS-assisted cystic duct cannulation was attempted in ten patients with high surgical risk and was successful in eight patients. 

• Adverse events: acute calculous cholecystitis recurrence and acute cholangitis with choledocholithiasis occurred during the follow-up period were not counted among the adverse events; authors might motivate this choice;

Thank you for your critical comments. I discussed about your comments with co-authors. We have concluded that it would be reasonable to analyze the adverse events including acute calculus cholecystitis recurrence and acute cholangitis. Thus, we changed our paper as follows: 

Adverse events occurred in 12.2% of patients (21/171: mild pancreatitis, n = 9; acute cholangitis, n=6; post-endoscopic sphincterotomy bleeding, n = 4; and stent migration, n = 1; ACC recurrence, n=1), but all patients were treated with conservative management and endoscopic treatment.

The delayed adverse events included asymptomatic complete stent migration, ACC recurrence, and acute cholangitis. Complete stent migration occurred in one patient at 93 days after ETGBS; however, there was no ACC recurrence until death due to pneumonia 61 days later. In one patient, ACC recurred at 43 days after the ETGBS, requiring an endoscopic reintervention. In six cases, acute cholangitis occurred at 150, 221, 299, 346, 381, and 399 days after the ETGBS and were managed with ERCP. 

Table 2. Outcomes of endoscopic transpapillary gallbladder drainage (n = 171)

Technical success rate, % (n) 90.6 (155$/171)

Clinical success rate (by ITT), % (n) 90.1 (154/171)

Adverse events 

Early adverse events 

 Pancreatitis, % (n) 5.2 (9)

 Post EST bleeding, % (n) 2.3 (4)

Late adverse events 

 Stent migration, % (n) 0.6 (1)

Acute calculus cholecystitis recurrence, % (n) 0.6 (1)

Acute cholangitis, % (n) 3.5 (6)

Procedure-related mortality, n 0

• Discussion: as reported by the authors, endoscopic transpapillary gallbladder drainage could be a challenging technique with some pitfalls and it sometimes requires a two-step approach; could the authors comment on the possibility of performing several endoscopic procedures on the same fragile patient rather than just one drainage session (e.g. EUS-GBD)?;

Morevoer, the Authors included in this study 171 patients. A definitive endoscopic treatment was performed only in 70 patients. This is the main difference between the transpapillary approach and the EUS-GBD. As reported in a recent review on gallbladder drainage(PMID 32523368 ) in patients with AC who are not candidates for surgery, EUS-GBD represents a more appropriate treatment compared to the transpapillary approach. ETGBD can still be considered in patients with choledocholithiasis, as it allows concomitant stone removal and physiologic drainage. Please comment this topic in the discussion including this review in the manuscript.

We already commented in the Discussion section as follows: “Recent studies comparing the EUS-GBD and ETGBD outcomes in patients with acute cholecystitis were published, and the authors reported that EUS-GBD resulted in higher technical and clinical success with lower cholecystitis recurrences than ETGBD.”

Considering your dedicated comments, we added the sentences as follows:

However, in high-volume centers with skilled interventional endoscopists, EUS-GBD is useful as a first line treatment modality for the management of ACC in patients with high surgical risk, especially in those with malignancies involving the cystic duct or an indwelled metal stent covered with a cystic duct orifice.[37] In addition, if ETGBS failed as an initial method for GB decompression, EUS-GBD may be considered as a secondary option in patients with ACC who are unfit urgent or interval cholecystectomy.

[37. Fugazza A, Colombo M, Repici A, Anderloni A. Endoscopic Ultrasound-Guided Gallbladder Drainage: Current Perspectives. Clin Exp Gastroenterol. 2020;13:193-201. doi: 10.2147/CEG.S203626. PubMed PMID: 32523368; PubMed Central PMCID: PMCPMC7237126.]

• Discussion: in the last part of the discussion it is said that some minor procedural complications may have been missed; could the authors be more specific?

This was a retrospective cohort study, therefore, it would be missed post procedural mild abdominal discomfort, which may be clinical insignificance, despite of careful prospective data collection. However, it would be possible that these sentences could bring several concerns or confusion to the readers. 

Thus we revised the sentences as follows: In this study, several limitations were noted. First, this was a retrospective study with possible selection bias and missed information. However, the data were prospectively collected and checked carefully.

Reviewer #2: This is a well designed retrospective analysis about endoscopic transpapillary GB drainage in acute calcolous colecistitis for patients unfit for urgent cholecistectomy.

In the LAMS (Lumen Apposing Metal stent) era it is crutial to discuss and evaluate an “anatomically natural” transpapillary approach in this setting of patients.

Please consider the following minor comments.

• In the Introduction section, following the sentence about LAMS “Moreover, technical failure and severe adverse events, such as gut perforation or bile peritonitis that may lead to sepsis or death, occur even in expert hands” I would add that in LAMS patients some concerns are raised in terms of feasibility and morbidity of interval cholecistectomy due to ahdesions between gallbladder and Stomach (or duodeum).

Because of that, if on one hand the EUS-guided gallbladder drainage with LAMS is gaining an important role for patients unfit for surgery, on the other hand it is still not proved it can be effectively used in patients who are possible candidates for delayed surgery

We added the sentences as follows: In addition, the feasibility and safety of interval cholecystectomy in patients who underwent EUS-GBD with a LAMS are not well reported, and there may be concerns due to severe inflammation or adhesion surrounding the gallbladder.

• In the material and Methods section, under the paragraph “endoscopic procedure and follow up” please edit the following sentence:

“In case of crossover from ENGBD to ETGBD, ENGBD catheter was removed […]” into the following “In case of crossover from ENGBD to ETGBS, ENGBD catheter was removed”

We revised our sentences as you mentioned. 

“In case of crossover from ENGBD to ETGBS, ENGBD catheter was removed following contrast injection through the catheter.” 

• About the patients who underwent SpyGlass assistance for cystic duct cannulation:

please specify the cases in whom Spy glass was used and if they were candidate for delayed surgery. 

Due to high costs this option maybe should be used in patient unfit for surgery with long stent/nasobiliary drainage indwelling time. 

In those patients for whom biliary drainage is a certain bridging option PT should be preferred in case of failure of standard cystic cannulation techniques

Even if you specified that a cost-effectiveness analysis has not been included in your study, a further description of this subpopulation of patients is beneficial to the clarity of the paper.

It may be hard to understand, but the Korean insurance system do not covered the cost of Spyglass DS system and we cannot also charge the patient for the Spyglass DS system. But, we used the SpyDS assisted cystic duct cannulation in patients with high surgical risk. 

We revised the sentences and added supplementary table as follows: 

In our study, Spy DS-assisted cystic duct cannulation was attempted in ten patients with high surgical risk and was successful in eight patients (Supplementary table 1). Therefore, the technical success rate increased by 4.6%.

Supplementary table 1. Baseline characteristics of patients with successful SpyDS assisted cystic duct cannulation. 

Case Age/Sex Co-morbid condition ASA class Diagnosis Visible cystic duct by cholangiography

1 87/F Complete AV block with pacemaker, cerebral infarction with left hemiparesis, HTN, DM III Presence of CBD stone with ACC No

2 80/F Dementia, heart failure, HTN III Presence of CBD stone with ACC No

3 83/F Parkinson’s disease, dilated cardiomyopathy, HTN, DM III Presence of CBD stone with ACC No

4 87/F Dementia, cerebral infarction, HTN, DM, CKD III Presence of CBD stone with ACC No

5 74/F Heart failure, mitral valve replacement, intracranial hemorrhage III Presence of CBD stone with ACC No

6 70/M ESRD with hemodialysis, DM, HTN III Presence of CBD stone with ACC No

7 79/F ESRD with hemodialysis, Parkinson’s disease, HTN III Presence of CBD stone with ACC No

8 81/F DM, HTN, dementia multiple myeloma III Presence of CBD stone with ACC No

SpyDS SpyglassTM DS Direct Visualization system; ASA, American Society of Anesthesiologists, CBD common bile duct, AV atrioventricular , ACC acute calculus cholecystitis, HTN hypertension, DM diabetes mellitus, CKD chronic kidney disease, ESRD end-stage renal disease.

---

## [Decision Letter · Decision Letter 1]

23 Sep 2020

Endoscopic transpapillary gallbladder drainage for the management of acute calculus cholecystitis patients unfit for urgent cholecystectomy

PONE-D-20-19852R1

Dear Dr. Chon,

We’re pleased to inform you that your manuscript has been judged scientifically suitable for publication and will be formally accepted for publication once it meets all outstanding technical requirements.

Kind regards,

Ezio Lanza, M.D.

Academic Editor

PLOS ONE

---

## [Editor Report · Acceptance letter]

28 Sep 2020

PONE-D-20-19852R1 

Endoscopic transpapillary gallbladder drainage for the management of acute calculus cholecystitis patients unfit for urgent cholecystectomy 

Dear Dr. Chon:

I'm pleased to inform you that your manuscript has been deemed suitable for publication in PLOS ONE. Congratulations! Your manuscript is now with our production department. 

Kind regards, 

on behalf of

Dr. Ezio Lanza 

Academic Editor

PLOS ONE